# Structured Federated Learning through Clustered Additive Modeling

**Jie MA[1], Tianyi Zhou[2], Guodong Long[1], Jing Jiang[1], Chengqi Zhang[1]**
[1]Australian Artificial Intelligence Institute, FEIT, University of Technology Sydney
[2]University of Maryland
jie.ma-5@student.uts.edu.au, tianyi@umd.au
{guodong.long, jing.jiang, chengqi.zhang}@uts.edu.au

## Abstract

Heterogeneous federated learning without assuming any structure is challenging due to the conflicts among non-identical data distributions of clients. In practice, clients often comprise near-homogeneous clusters so training a server-side model per cluster mitigates the conflicts. However, FL with client clustering often suffers from "clustering collapse", i.e., one cluster's model excels on increasing clients, and reduces to single-model FL. Moreover, cluster-wise models hinder knowledge sharing between clusters and each model depends on fewer clients. Furthermore, the static clustering assumption on data may not hold for dynamically changing models, which are sensitive to cluster imbalance/initialization or outliers. To address these challenges, we propose "**C**lustered **A**dditive **M**odeling (**CAM**)", which applies a globally shared model $\Theta_g$ on top of the cluster-wise models $\Theta_{1:K}$, i.e., $y = h(x; \Theta_g) + f(x; \Theta_k)$ for clients of cluster-$k$. The global model captures the features shared by all clusters so $\Theta_{1:K}$ are enforced to focus on the difference among clusters. To train CAM, we develop a novel **Fed-CAM** algorithm that alternates between client clustering and training global/cluster models to predict the residual of each other. We can easily modify any existing clustered FL methods by CAM and significantly improve their performance without "clustering collapse" in different non-IID settings. We also provide a convergence analysis of Fed-CAM algorithm.

## 1 Introduction

Federated learning (FL) trains a global model over distributed clients and enforces data localization, i.e., data only stay local for model training at the client side while the server periodically averages client models' weights to update a global model and broadcast it to all clients. When the local data distributions are identical across clients, one global model suffices to serve all clients [1] However, non-identical distributions across clients (i.e., statistical heterogeneity) [2] are more common in practical FL scenarios, which leads to conflicts between the global objective and local ones. Instead of applying one model to all the $m$ clients, an ideal case for non-IID settings would be training a local model per client without any interference from others. However, the local data are usually insufficient so a global model trained on heterogeneous clients exploiting all their data can still be helpful to local training. Hence, non-IID FL methods [3–5] usually struggle to find a sweet spot between the global consensus and local personalization. Without any assumptions on the structure among clients, all clients' distributions can be equally different from each other so a global model is impacted by the conflicts of all clients and may provide limited guidance to their local training.

**Clustered Federated Learning.** That being said, non-IID clients in practice usually have rich structures that have not been explored by most existing FL methods. A common structure is clusters, i.e., heterogeneous clients can be grouped into several near-homogeneous clusters each composed of

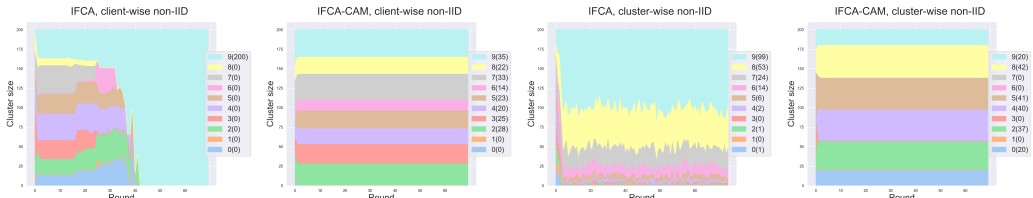

Figure 1: Cluster sizes during IFCA vs. IFCA+CAM in client/cluster-wise non-IID settings on CIFAR-10. Legend: cluster ID (cluster size) in the last round. **CAM effectively mitigates clustering collapse/imbalance.**

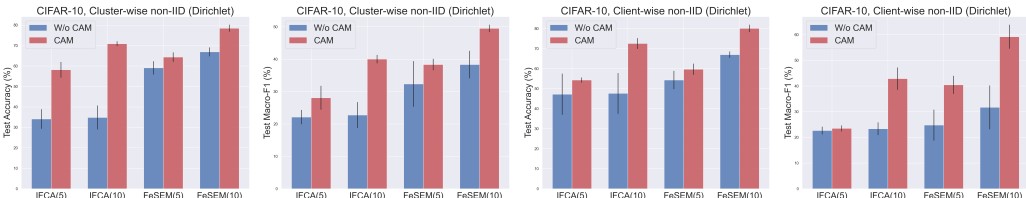

Figure 2: Test accuracy and macro-F1 (mean±std) of IFCA/FeSEM (w/o CAM) and IFCA/FeSEM (CAM) in cluster/client-wise non-IID settings on CIFAR-10 dataset. "IFCA(5)" represents IFCA with $K = 5$ clusters. **CAM consistently brings substantial improvement to IFCA/FeSEM on both metrics and in both settings.**

clients with similar distributions. In practice, clusters might be associated with geographic/age/income groups, affiliations, etc. Hence, we can train a server-side model for each cluster, hence mitigating the conflicts caused by heterogeneity. Unfortunately, clients' cluster memberships are usually undefined or inaccessible due to sensitive/private information and have to be jointly optimized with cluster-wise models, as recent clustered FL [6–9] approaches do. They maintain $K$ models $\Theta_{1:K}$ for $K$ clusters and assigns one $\Theta_k$ to each client-$i$ (with local data $X_i$ and local model $\theta_i$), e.g., by min-loss ($\Theta_k$ with the minimum loss on $X_i$) or K-means (the nearest $\Theta_k$ to $\theta_i$) criterion. Hence, $1 \leq K \leq m$ models can accommodate more heterogeneity than single-model FL but also allows knowledge sharing among similar clients, which is lacking if training $m$ client models independently. Hence, it may reach a better trade-off between global consensus and local personalization in non-IID settings.

However, compared to the general non-IID assumption, *clustered FL's assumption might be too restrictive since it prohibits inter-cluster knowledge sharing and enforces every cluster-wise model's training to only depend on a few clients.* This is contradictory to the widely studied strategy that different tasks or domains can benefit from sharing low-level or partial representations. It is due to the gap between the assumption of "clustered data distributions" and the algorithms of "clustering models (represented by loss vectors or model weights)": they are not equal and the latter is more restrictive. In other words, clients of different clusters can still benefit from feature/parameter sharing.

Moreover, *clustered FL usually suffers from optimization instability because dynamically changing models* can violate the static clustering assumption and lead to imbalanced cluster assignment, which affects $\Theta_{1:K}$ and local training in the future. In particular: (1) *Clustering collapse*, i.e., the clients assigned to one cluster keeps growing so "the rich becomes richer (i.e., the cluster-wise model becomes even stronger)" until reducing to single-model FL. This happens because most clients tend to first learn shared features before focusing on client-specific ones; (2) *Fragile to outliers* such as malicious clients that may dominate some clusters and push all other benign ones to one or a few clusters; (3) *Sensitive to initialization.* The process highly depends on initial and earlier cluster assignments since they determine which clients' local training starts from the same model.

**Main Contributions.** To overcome the above problems of clustered FL, we propose a novel structured FL model termed "**C**lustered **A**dditive **M**odeling (**CAM**)". Compared to clustered FL, CAM trains a global model $\Theta_g$ on top of the $K$ clusters' models $\Theta_{1:K}$. Its prediction for client-$i$ combines the outputs of $\Theta_g$ and the associated cluster $c(i)$'s model, i.e., $y = h(x; \Theta_g) + f(x; \Theta_{c(i)})$. This simple additive model removes the restriction of clustered FL by letting all clients share a base model $\Theta_g$. It enforces $\Theta_{1:K}$ to focus on learning the different features between clusters, hence mitigating "clustering collapse". Moreover, CAM tends to learn balanced clusters and determine the number of clusters automatically (by starting from more clusters and then zeroing out some of them), as demonstrated in Fig 1. Furthermore, CAM is less vulnerable to outliers, which can be mainly captured by $\Theta_{1:K}$ and have less impact on the global model $\Theta_g$. In addition, interactions between $\Theta_{1:K}$ and $\Theta_g$ make CAM less sensitive to initial cluster assignments since updating $\Theta_g$ also changes the clustering.

CAM is a general model-agnostic method that can modify any existing clustered FL methods. As examples, we apply CAM to two representative methods, i.e., IFCA [6] and FeSEM [8]. In the optimization of CAM, $\Theta_{1:K}$ and $\Theta_g$ aim to fit the residual of each other's prediction. To this end, we propose an efficiently structured FL algorithm "**Fed-CAM**", which alternates between cluster assignment (server), local training (clients), and update of $\Theta_{1:K}$ and $\Theta_g$ (server). In experiments on several benchmarks in different non-IID settings, CAM significantly improves SOTA clustered FL methods, as shown in Fig 2. Moreover, we provide a convergence analysis of Fed-CAM algorithm.

## 2 Related Work

**Non-IID FL** aims to tackle statistical heterogeneity across clients. FedAvg [1] is designed for the IID setting, so it suffers from client drift and slow convergence with non-IID clients [2]. To address this challenge, FedDANE [10] proposed a federated Newton-type optimization method by adapting a method for classical distributed optimization, i.e., DANE, to the FL setting. Instead of synchronizing all clients' models to be the same global model periodically, FedProx [3] only adds a proximal term to the local training objective that discourages the local model from drifting away from the global model and thus preserves the heterogeneity. [11] applies adaptive learning rates to clients and [12] conducts attention-based adaptive weighting to aggregate clients' models. [13] studies the convergence of the FedAvg in non-IID scenarios. Recent work also studies client-wise personalized FL [14–20], which aim to address the non-IID challenge by training a personalized model per client with the help of the shared global model. Their objectives focus on training local models rather than the server-side model.

**Clustered FL** assumes that non-IID clients can be partitioned into several groups and clients in each group share a cluster-wise model. It jointly optimizes the cluster assignments and the clusters' models. K-means-based methods [8] assign clusters to clients according to their model parameters' distance. CFL [21] divides clients into two partitions based on the cosine similarity between client gradients and then checks whether a partition is congruent according to the gradient norm. IFCA [6] and HypCluster [7] assign to each client the cluster whose model achieves the minimum loss on the client's data. Few-shot clustering has been introduced to clustered FL by [22,23]. FedP2P [24] allows communication between clients in the same cluster. [25] uses cluster-based contexts to enhance the fine-tuning of personalized FL models. [9] proposed the first cluster-wise non-IID setting and a bi-level optimization framework unifying most clustered FL methods.

**Additive modeling in FL** trains multiple models and adds their outputs together as its prediction. It was introduced to FL very recently. Federated residual learning [26] proposed an FL algorithm to train an additive model for regression tasks. [27] applies additive modeling to combining the outputs of a shared model and a local model in a partial model personalization framework. However, additive modeling has not been studied for clustered FL.

## 3 Clustered Additive Modeling (CAM)

In this section, we introduce clustered additive modeling (CAM), which combines a global model and cluster-wise model prediction in FL. CAM conducts a joint optimization of the global and cluster-wise models defined by a cluster assignment criterion. In particular, we provide two examples of CAM using different cluster assignment criteria, i.e., min-loss and K-means, which have been adopted respectively by two SOTA clustered-FL methods, i.e., IFCA and FeSEM. For each of them, we derive alternating optimization procedures (i.e., IFCA-CAM and FeSEM-CAM) that can be implemented in FL setting using two parallel threads of local model training. At the end of this section, we unify both algorithms in a structured FL algorithm Fed-CAM.

**Notations.** We assume that there are $m$ clients and $K$ clusters, where client-$i$ has $n_i$ examples and all clients have $n = \sum_{i=1}^{m} n_i$ examples. On the server side, we have a global model $\Theta_g$ and $K$ cluster-wise models $\Theta_{1:K}$. On the client side, we train $m$ cluster models $\theta_{1:m}^0$ used to update the global model $\Theta_g$ in FL and $\theta_{1:m}$ used to update the cluster-wise model $\Theta_{c(i)}$ assigned to each client-$i$, where $c(i)$ is its cluster label determined by the cluster assignment criterion $c(\cdot)$. We further define $C_k \triangleq \{i \in [m] : c(i) = k\}$ as the set of clients in cluster-$k$. For simplicity, we will use $X_i$ and $Y_i$ to respectively represent the local training data on client-$i$ and their ground truths, and $\ell(Y_i, F(X_i))$ denotes the batch loss of model $F(\cdot)$ on $(X_i, Y_i)$. A CAM model for client-$i$ can be

$$F_i(\cdot) = h(\cdot; \Theta_g) + f(\cdot; \Theta_{c(i)}). \tag{1}$$

For classification, $F_i(\cdot)$ produces logits and we can apply softmax to get the class probabilities.

## 3.1 IFCA-CAM: model performance-driven clustering

We extend the min-loss criterion used in IFCA [6] to CAM for cluster assignment, i.e., each client-$i$ is assigned to the cluster-$k$ whose model $\Theta_k$ leads to the minimal loss of CAM on client-$i$'s data, i.e.,

$$c(i) = \arg \min_{k \in [K]} \ell(Y_i, h(X_i; \Theta_g) + f(X_i; \Theta_k)). \tag{2}$$

IFCA-CAM optimizes $\Theta_g$ and $\Theta_{1:K}$ for minimizing the above minimal loss over all the $m$ clients, i.e.,

$$\text{IFCA-CAM:} \quad \min_{\Theta_g, \Theta_{1:K}} \sum_{i=1}^{m} \frac{n_i}{n} \min_{k \in [K]} \ell(Y_i, h(X_i; \Theta_g) + f(X_i; \Theta_k)), \tag{3}$$

where the inner minimization performs the min-loss assignment in Eq. (2). We solve Eq. (3) by the following alternating minimization of cluster membership, cluster-wise models, and the global model.

**(i)** Cluster assignment by applying Eq. (2) to the latest $\Theta_g$ and $\Theta_{1:K}$. This yields $c(\cdot)$ and $C_{1:K}$.

**(ii)** Fixing $\Theta_g$, we can optimize the cluster-wise models $\Theta_{1:K}$ by gradient descent:

$$\Theta_k \leftarrow \Theta_k - \eta \sum_{i \in C_k} \frac{n_i}{n} \nabla_{\Theta_k} \ell(Y_i, h(X_i; \Theta_g) + f(X_i; \Theta_k)), \ \forall \, k \in [K]. \tag{4}$$

In FL, the gradient can be approximated by aggregating the model updates of local models $\theta_i$ from clients, whose training on the client side is: (1) initializing $\theta_i \leftarrow \Theta_{c(i)}$; (2) starting from the initialization, running $E$ local epochs updating $\theta_i$ by

$$\theta_i \leftarrow \theta_i - \eta \nabla_{\theta_i} \ell(Y_i, h(X_i; \Theta_g) + f(X_i; \theta_i)), \ \forall \, i \in [m]; \tag{5}$$

and (3) aggregating the local model update $\theta_i - \Theta_k$ from client $i \in C_k$ to update $\Theta_k$, i.e.,

$$\Theta_k \leftarrow \left( 1 - \sum_{i \in C_k} \frac{n_i}{n} \right) \Theta_k + \sum_{i \in C_k} \frac{n_i}{n} \theta_i. \tag{6}$$

**(iii)** Fixing $\Theta_{1:K}$, we can optimize the global model $\Theta_g$ by gradient descent:

$$\Theta_g \leftarrow \Theta_g - \eta \sum_{i \in [m]} \frac{n_i}{n} \nabla_{\Theta_g} \ell(Y_i, h(X_i; \Theta_g) + f(X_i; \Theta_{c(i)})). \tag{7}$$

In FL, this gradient step can be approximated by aggregating the local models $\theta_i^0$ (similar to FedAvg): (1) initializing $\theta_i^0 \leftarrow \Theta_g$; (2) running $E$ local epochs training $\theta_i^0$ by

$$\theta_i^0 \leftarrow \theta_i^0 - \eta \nabla_{\theta_i^0} \ell(Y_i, h(X_i; \theta_i^0) + f(X_i; \Theta_{c(i)})), \ \forall \, i \in [m]; \tag{8}$$

and (3) aggregating the updated local models $\theta_i^0$ of all the $m$ clients to update $\Theta_g$, i.e.,

$$\Theta_g \leftarrow \sum_{i \in [m]} \frac{n_i}{n} \theta_i^0. \tag{9}$$

We can run two parallel threads of local training for $\theta_i^0$ and $\theta_i$ for each client-$i$ because their training in Eq. (8) and Eq. (5) does not depend on each other (but they both depend on the cluster assignments in (i)). This is analogous to the simultaneous update algorithm (FedSim) in [27]. One may also consider an alternative update algorithm (which may enjoy a slightly faster convergence) that iterates (i)→(ii)→(i)→(iii) in each round. However, it doubles the communication rounds ((i) requires one communication round) and does not allow parallel local training. Since the alternative update does not show a significant empirical improvement over FedSim in [27], we mainly focus on the parallel one in the remainder of this paper.

## 3.2 FeSEM-CAM: parameter similarity-based clustering

We follow a similar procedure of IFCA-CAM to derive FeSEM-CAM, which applies a K-means style clustering to the client models $\theta_{1:m}$, whose objective is minimizing the sum of squares of client-cluster distance, i.e.,

$$\min_{\Theta_{1:K}} \sum_{i=1}^{m} \frac{n_i}{n} \min_{j \in [K]} \|\theta_i - \Theta_j\|_2^2. \tag{10}$$

Hence, similar to FeSEM [8], FeSEM-CAM assigns the nearest cluster-wise model to each client and updates the cluster-wise models as the cluster centroids (i.e., K-means algorithm), i.e.,

$$c(i) = \arg \min_{k \in [K]} \|\theta_i - \Theta_k\|_2^2, \ \ \Theta_k \leftarrow \sum_{i \in C_k} \frac{n_i}{\sum_{i \in C_k} n_i} \theta_i. \tag{11}$$

We iterate the above K-means steps for a few times until convergence in practice. FeSEM-CAM applies the K-means objective in Eq. (10) as a regularization to the loss of CAM model $\ell(Y_i, h(X_i; \Theta_g) + f(X_i; \theta_i))$, i.e.,

$$\text{FeSEM-CAM:} \quad \min_{\Theta_g, \Theta_{1:K}, \theta_{1:m}} \sum_{i=1}^{m} \frac{n_i}{n} \left[ \ell(Y_i, h(X_i; \Theta_g) + f(X_i; \theta_i)) + \frac{\lambda}{2} \min_{j \in [K]} \|\theta_i - \Theta_j\|_2^2 \right], \tag{12}$$

where the minimization w.r.t. $\Theta_{1:K}$ (with $\theta_{1:m}$ fixed) recovers the (weighted) K-means objective in Eq. (10). Unlike IFCA-CAM, where client model $\theta_i$ is an auxiliary/latent variable for FL not showing in the objective of Eq. (3), it is explicitly optimized in Eq. (12). Similar to IFCA-CAM, we solve Eq. (3) by iterating the following alternating minimization steps (i)-(iii).

**(i)** K-means clustering that iterates Eq. (11) for a few steps until convergence, which yields $c(\cdot)$, $C_{1:K}$, and $\Theta_{1:K}$. The update of $\Theta_{1:K}$ is analogous to Eq. (6).

**(ii)** Fixing $\Theta_g$, we optimize $\theta_{1:m}$ by client-side local gradient descent:

$$\theta_i \leftarrow (1 - \eta\lambda)\theta_i + \eta\lambda\Theta_{c(i)} - \eta\frac{n_i}{n}\nabla_{\theta_i}\ell(Y_i, h(X_i; \Theta_g) + f(X_i; \theta_i)), \ \forall i \in [m]. \tag{13}$$

The first two terms in Eq. (13) compute a linear interpolation between $\theta_i$ and its assigned cluster's model $\Theta_{c(i)}$. This is a result of the K-means regularization term in Eq. (12) and keeps $\theta_i$ close to $\Theta_{c(i)}$.

**(iii)** Fixing $\theta_{1:m}$, we can optimize $\Theta_g$ by gradient descent:

$$\Theta_g \leftarrow \Theta_g - \eta \sum_{i \in [m]} \frac{n_i}{n} \nabla_{\Theta_g} \ell(Y_i, h(X_i; \Theta_g) + f(X_i; \theta_i)). \tag{14}$$

In FL, this gradient step can be approximated by aggregating the local models $\theta_i^0$ (similar to FedAvg): (1) initializing $\theta_i^0 \leftarrow \Theta_g$; (2) running $E$ local epochs training $\theta_i^0$ by

$$\theta_i^0 \leftarrow \theta_i^0 - \eta\nabla_{\theta_i^0}\ell(Y_i, h(X_i; \theta_i^0) + f(X_i; \theta_i)), \ \forall i \in [m]; \tag{15}$$

and (3) aggregating the updated local models $\theta_i^0$ of all the $m$ clients to update $\Theta_g$ by Eq. (9).

### 3.3 Algorithm

In Algorithm 1, we propose a structured FL algorithm for CAM, i.e., Fed-CAM, which can unify the derived optimization procedures for IFCA-CAM and FeSEM-CAM and can be easily extended to other clustered FL and clustering criteria.

**Warmup.** As an alternating optimization framework, it would be unstable if both $\Theta_g$ and $\Theta_{1:K}$ are randomly initialized and jointly optimized in parallel since they may capture overlapping information and result in an inefficient competitive game. To encourage them to learn complementary knowledge, warmup training for one of them before the joint optimization is helpful. For example, a few rounds of FedAvg can produce a "warm" $\Theta_g$, whose predictions' residuals are more informative to train $\Theta_{1:K}$. Another warmup strategy could be to run a few local training epochs and extract warm $\Theta_{1:K}$ by clustering the lightly-trained local models $\theta_{1:m}$. In Fed-CAM, we can apply the former warmup to IFCA-CAM and the latter to FeSEM-CAM.

**Algorithm 1:** Fed-CAM

**initialize** :Randomly initialize $\Theta_{1:K}$ and $\Theta_g$.
**warmup** :(1) $\beta$ rounds of FedAvg to get an initial $\Theta_g$ (IFCA-CAM) or (2) $\beta$ epochs of local training only to get a initial $\theta_{1:m}$ (FedSEM-CAM). Broadcast $\Theta_g$ and $\Theta_{1:K}$ to all clients.

**1 while** *not converge* **do**
  /* Client (in parallel)                         */
**2** **for** *every selected client $i$* **do**
**3**   Model performance-driven clustering (e.g., IFCA-CAM): cluster assignment by Eq. (2);
**4**   Initialize $\theta_i \leftarrow \Theta_{c(i)}$ and $\theta_i^0 \leftarrow \Theta_g$;
**5**   Local training of $\theta_i$ for $E$ epochs: e.g., Eq. (5) (IFCA-CAM) or Eq. (13) (FeSEM-CAM);
**6**   Local training of $\theta_i^0$ for $E$ epochs: e.g., Eq. (8) (IFCA-CAM) or Eq. (15) (FeSEM-CAM);
**7**   Upload $\theta_i$ and $\theta_i^0$ to the server;
  /* Server                                */
**8** Update cluster-wise models $\Theta_{1:K}$: e.g., Eq. (6) (IFCA-CAM) or Eq. (11) (FeSEM-CAM);
**9** Update global model $\Theta_g$ by Eq. (9);
**10** Broadcast $\Theta_g$ and $\Theta_{1:K}$ to all clients;

**output** :Global model $\Theta_g$, cluster-wise models $\Theta_{1:K}$ and $c(i) \, \forall \, i \in [m]$.

## 4 Convergence Analysis

Based on the convergence analysis presented in [27], which aims to minimize the following objective:

$$\min_{u,V} F(u,V) := \frac{1}{n} \sum_{i=1}^{m} F_i(u, v_i), \tag{16}$$

where $u$ represents the shared parameters and $V = v_1, v_2, \cdots, v_m$ denotes the personalized parameters. If we map $\Theta_g$ to $u$, and $\Theta_{1:K}$ to $V$ respectively, this appears strikingly similar to our methods as illustrated in Equations 3 and 12. Provided that the clustering remains stable, we can employ the theoretical framework of [27]. And firstly, we make some standard assumptions for the convergence analysis as below.

**Assumption 1.** (Smoothness). For $i = 1, \cdots, m$, the loss function $l$ is continuously differentiable, and there exist constants $L$ that $\nabla_{\Theta_g}\ell(\Theta_g, \Theta_k)$ is L-Lipschitz with respect to $\Theta_g$ and $\Theta_k$, and $\nabla_{\Theta_k}\ell(\Theta_g, \Theta_k)$ is L-Lipschitz with respect to $\Theta_g$ and $\Theta_k$.

**Assumption 2.** (Unbiased gradients and bounded variance). The stochastic gradients are unbiased and have bounded variance. For all $\Theta_g$ and $\Theta_k$,

$$\mathbb{E}[\widetilde{\nabla}_{\Theta_g}\ell(\Theta_g, \Theta_k)] = \nabla_{\Theta_g}\ell(\Theta_g, \Theta_k), \ \ \mathbb{E}[\widetilde{\nabla}_{\Theta_k}\ell(\Theta_g, \Theta_k)] = \nabla_{\Theta_k}\ell(\Theta_g, \Theta_k),$$

and

$$\mathbb{E}[\|\widetilde{\nabla}_{\Theta_g}\ell(\Theta_g, \Theta_k) - \nabla_{\Theta_g}\ell(\Theta_g, \Theta_k)\|^2] \leq \sigma_g^2, \ \ \mathbb{E}[\|\widetilde{\nabla}_{\Theta_k}\ell(\Theta_g, \Theta_k) - \nabla_{\Theta_k}\ell(\Theta_g, \Theta_k)\|^2] \leq \sigma_k^2.$$

**Assumption 3.** (Partial gradient diversity). There exists a constant for all $\theta_i^0$ and $\Theta_g$, $\theta_i$ and $\Theta_k$,

$$\sum_{i=1}^{m} \frac{n_i}{n} \|\nabla_{\Theta_g}\ell(\Theta_g, \theta_i) - \nabla_{\Theta_g}\ell(\Theta_g, \Theta_k)\|^2 \leq \delta^2$$

$$\sum_{i \in [k]} \frac{n_i}{\sum_{i \in [k]} n_i} \|\nabla_{\Theta_k}\ell(\theta_i^0, \Theta_k) - \nabla_{\Theta_k}\ell(\Theta_g, \Theta_k)\|^2 \leq \delta^2.$$

**Assumption 4.** (Convexity of cluster models). Fix $\Theta_g$, assume $\ell(\Theta_k)$ is convex.

**Theorem 1.** *(Convergence of Fed-CAM). Let Assumptions 1, 2, 3 and 4 hold, and learning rates chosen as $\eta = \tau/(LE)$ for a $\tau$ depending on the parameters $L, \sigma_g^2, \sigma_k^2, \delta^2, s, m, T$, provided clustering stable, we have (ignoring absolute constants),*

$$\frac{1}{T} \sum_{t=1}^{T} \left( \frac{1}{L} \mathbb{E}[\|\nabla_{\Theta_g}\ell(\Theta_g, \Theta_k)\|^2] + \frac{s}{mL} \frac{1}{m} \sum_{i=1}^{m} \mathbb{E}[\|\nabla_{\Theta_{c(i)}}\ell(\Theta_g, \Theta_{c(i)})\|^2] \right) \tag{17}$$

$$\leq \frac{(\triangle_\ell \sigma_{sim,1}^2)^{1/2}}{T^{1/2}} + \frac{(\triangle_\ell^2 \sigma_{sim,2}^2)^{1/3}}{T^{2/3}} + O(\frac{1}{T}), \tag{18}$$

where $\triangle_\ell = \ell_0 - \ell^\star$, and we define effective variance terms,

$$\sigma_{sim,1}^2 = \frac{2}{L}(\delta^2(1 - \frac{s}{m}) + \frac{\sigma_g^2}{L} + \frac{\sigma_k^2 s}{m})) \tag{19}$$

$$\sigma_{sim,2}^2 = \frac{2}{L}(\delta^2 + \sigma_g^2 + \sigma_k^2)(1 - \frac{1}{E}). \tag{20}$$

*Remark* 1. It is straightforward to prove that the clustering of both IFCA-CAM and FeSEM-CAM converges, as evidenced by Ma et al. (2022). However, proving the stability of these clustering methods is more challenging due to the oscillation phenomenon often seen in K-means. The stability of clustering will be further demonstrated through experimental analysis in Section 5.2.

*Remark* 2. Besides the clustering structure, there is a distinct difference between FedSim [27] and Fed-CAM. In Fed-CAM, we need to aggregate both $\Theta_g$ and $\Theta_{1:K}$, while in FedAlt [27], only $\Theta_g$ requires aggregation. The $\sigma_{sim,1}^2$ and $\sigma_{sim,2}^2$ reflect the impact of sample number $s$ and local steps $E$. Larger $s$ or smaller $E$, better convergence rate. According to the results presented in [27], alternative gradient descent surpasses simultaneous gradient descent in terms of convergence rate. The asymptotic $1/\sqrt{T}$ rate is achieved when each device is seen at least once on average, and the $1/T$ term is dominated by the $1/\sqrt{T}$ term, a situation that occurs when (ignoring absolute constants)

$$T \geq \frac{\triangle_\ell}{\sigma_{sim,1}^2} \max\{(1 - \frac{1}{E})\frac{m}{s}, 2\}.$$

## 5  Experiments

**Benchmark datasets and partitions.** The proposed methods have been validated using several benchmark datasets. While detailed results for PathMNIST and TissueMNIST from the MedMNIST [28] are provided in the supplementary material, the other datasets used for validation include:

- **Fashion-MNIST** [29] includes 70,000 labeled fashion images (28×28 grayscale) in 10 classes, such as T-shirts, Trouser, and Bag, with others.
- **CIFAR-10** [30] consists of 60,000 images (32×32 color) in 10 classes, including airplane, automobile, bird, and truck, among others. The divergence among classes in CIFAR-10 is relatively higher than in other datasets from the MNIST family.

Each dataset is split among 200 clients and we create the following non-IID scenarios:

- **Client-wise non-IID by Dirichlet distribution ($\alpha = 0.1$)**: This approach uses the Dirichlet distribution to control the randomness of non-IID data, as proposed by [31]. This is a standard method used in most personalized FL methods, which are usually client-wise non-IID.
- **Cluster-wise non-IID by Dirichlet distribution ($\alpha = (0.1, 10)$)**: This strategy divides the dataset into $K$ clusters with $\alpha = 0.1$ to generate substantial variance in cluster-wise non-IID. Then, each cluster is divided into $m/K$ clients with $\alpha = 10$ to control the non-IID across clients.
- **Client-wise non-IID by n-class (2)**: This method randomly selects $n$-class out of all classes in the dataset for each client, as proposed by [1], and then samples instances from these datasets.
- **Cluster-wise non-IID by n-class (3, 2)**: This approach randomly assigns 3 classes to each cluster, ensuring a relatively balanced number of instances per class. It then assigns 2 classes to each client.

**Baselines.** We select baseline methods from four categories as follows:

- **Single model-based FL:** We choose FedAvg [1] and FedProx [3] with a coefficient of 230 and a regularization of 0.95 as the baselines.
- **Ensemble FL:** We train FedAvg and FedProx $K$ times and then learn an ensemble model via soft voting to serve all clients, which are named FedAvg+ and FedProx+, respectively.
- **Clustered FL:** We choose FeSEM [8] and IFCA [6], which is similar to HypCluster [7].
- **Clustered FL with additive modeling:** We integrate CAM with IFCA and FeSEM, denoting them as IFCA-CAM and FeSEM-CAM, respectively.

**Learning-related hyperparameters.** We use the Convolutional Neural Network (CNN) [32] as the basic model architecture for each client, as detailed in the supplementary material. For optimization, we employ SGD with a learning rate of 0.001 and momentum of 0.9 to train the model, and the batch size is 32. We evaluate the performance using both **micro accuracy** (%) and **macro F1-score** (%) on the client-wise test datasets to better capture the non-IID nature per client.

**FL system settings.** We conduct 100 global communication rounds in the FL system, including 30 warm-up rounds if applicable. Each communication involves 10 local steps. For the clustering process of FeSEM-CAM, we measure distance on the flattened parameters of the fully-connected layers, and use K-Means as the clustering algorithm. The coefficient $\lambda$ is chosen from $0.001, 0.01, 0.1$ based on the best performance.

## 5.1 Main Results and Comparisons

**Cluster-wise non-IID** scenarios make the assumption that there are underlying clustering structures among clients. Table 1 compares the methods using two benchmark datasets, namely Fashion-MNIST and CIFAR-10. Results using two biomedical datasets are presented in the appendix. The following are some notable observations and analyses:

- The application of the ensemble mechanism to FedAvg and FedProx yields minor improvements. This is because the server-side model in FedAvg or FedProx is already a relatively strong model, while ensemble mechanisms usually excel with weak models.

- The introduction of CAM significantly enhances the performance of IFCA, which typically struggles with clustering collapse in cluster-wise non-IID scenarios. Notably, CAM decomposes the shared components into a global model and personalized parts into cluster models. Thus, the clustering collapse is mitigated by isolating the dominant shared knowledge.

- FeSEM generally exhibits robust performance on cluster-wise non-IID without outliers. Implementing CAM in FeSEM further improves the Macro-F1 performance. The clustering process in FeSEM tends to overfit the label distribution (imbalanced classes) of clients to achieve higher accuracy. However, the application of CAM introduces a global model with a balanced label distribution by averaging all clients, thereby boosting the Macro-F1 performance while preserving the cluster-wise non-IID for high accuracy.

- With an increase in the number of clusters $K$, the CAM-based methods show substantial improvements in Macro-F1. The decomposition of shared knowledge and cluster-wise non-IID characteristics benefit from a reasonably larger $K$, which facilitates fine-grained, cluster-wise personalization.

Table 1: Test results (mean±std) in **cluster**-wise non-IID settings on Fashion-MNIST & CIFAR-10.

| Datasets | | Fashion-MNIST | | | | CIFAR-10 | | | |
| --- | --- | --- | --- | --- | --- | --- | --- | --- | --- |
| Non-IID setting | | Dirichlet $\alpha = (0.1, 10)$ | | n-class $(3, 2)$ | | Dirichlet $\alpha = (0.1, 10)$ | | n-class $(3, 2)$ | |
| #Cluster | Methods | Accuracy | Macro-F1 | Accuracy | Macro-F1 | Accuracy | Macro-F1 | Accuracy | Macro-F1 |
| **1** | FedAvg | 86.08±0.70 | 57.24±2.26 | 86.33±0.44 | 46.09±1.08 | 24.38±3.30 | 11.69±3.15 | 21.33±3.83 | 9.00±0.58 |
| | FedProx | 86.32±0.78 | 58.03±3.19 | 86.42±0.63 | 45.86±1.42 | 24.73±3.68 | 11.28±2.35 | 22.66±1.13 | 9.23±0.78 |
| **5** | FedAvg+ | 87.61 | 59.48 | 86.95 | 65.61 | 25.97 | 12.16 | 24.35 | 9.06 |
| | FedProx+ | 87.94 | 59.83 | 86.52 | 65.73 | 26.05 | 12.53 | 24.83 | 9.31 |
| | IFCA | 84.60±2.22 | 62.03±3.01 | 84.94±2.54 | 66.50±4.43 | 34.10±4.79 | 22.12±2.21 | 29.80±4.49 | 17.90±2.08 |
| | IFCA-CAM | 93.33±0.95 | 79.64±4.09 | 95.38±0.49 | 77.56±1.14 | 58.13±3.82 | 28.09±3.68 | 54.56±3.58 | 27.27±1.06 |
| | FeSEM | 94.64±1.54 | 82.90±2.38 | 94.20±1.96 | 77.07±6.05 | 59.06±3.24 | 32.33±7.25 | 58.76±3.35 | 35.75±2.54 |
| | FeSEM-CAM | **95.13±1.78** | **85.10±3.17** | **95.69±1.05** | **78.82±1.17** | **64.35±2.33** | **38.33±1.77** | **65.58±1.21** | **38.63±1.17** |
| **10** | FedAvg+ | 89.42 | 67.83 | 86.91 | 63.01 | 28.45 | 13.79 | 27.28 | 9.81 |
| | FedProx+ | 89.55 | 68.02 | 86.73 | 63.42 | 28.33 | 13.64 | 26.94 | 9.64 |
| | IFCA | 82.10±5.40 | 62.62±8.22 | 86.58±4.97 | 66.22±5.69 | 34.84±5.82 | 22.76±3.99 | 34.06±2.60 | 18.70±1.31 |
| | IFCA-CAM | 95.42±2.54 | 88.45±5.46 | 95.09±0.87 | 82.98±1.16 | 70.90±1.18 | 40.03±1.28 | 68.46±4.08 | 41.45±4.00 |
| | FeSEM | 95.73±1.28 | 89.34±1.57 | 95.54±0.74 | 84.43±2.38 | 66.89±2.18 | 38.35±4.24 | 71.76±2.23 | 49.72±3.84 |
| | FeSEM-CAM | **96.19±1.20** | **92.37±1.85** | **98.07±1.46** | **92.43±2.70** | **78.45±1.71** | **49.50±1.13** | **75.04±1.97** | **55.90±2.07** |

**Client-wise non-IID** Table 2 presents comparative results under client-wise non-IID scenarios using two benchmark datasets: Fashion-MNIST and CIFAR-10. Interestingly, IFCA maintains stable performance under client-wise non-IID conditions, primarily because it cannot form a single dominant cluster model - a primary cause of clustering collapse - in a highly heterogeneous environment. The application of CAM to IFCA and FeSEM shows a significant enhancement, particularly on the CIFAR-10 dataset. This improvement is likely due to FeSEM's typical restriction on knowledge sharing across clusters. In contrast, CAM utilizes a global model to capture more useful common

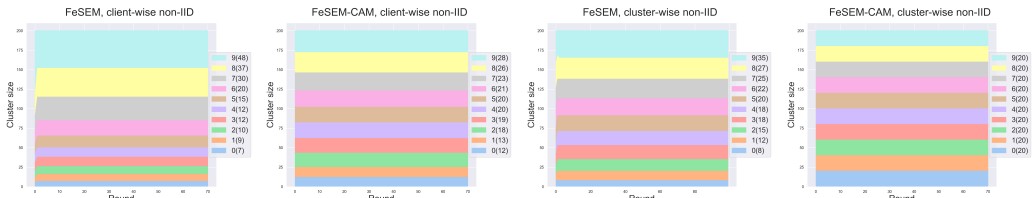

Figure 3: Cluster sizes during FeSEM vs. FeSEM+CAM in client/cluster-wise non-IID settings on CIFAR-10. Legend: cluster ID (cluster size) in the last round. **CAM effectively mitigates clustering collapse/imbalance**.

knowledge across clusters, thereby substantially enhancing the generalization capability of each cluster. Furthermore, CIFAR-10, being a relatively complex dataset with a diversity of images, underscores the importance of sharing common knowledge.

Table 2: Test results (mean±std) in **client**-wise non-IID settings on Fashion-MNIST & CIFAR-10.

| Datasets | | Fashion-MNIST | | | | CIFAR-10 | | | |
|---|---|---|---|---|---|---|---|---|---|
| Non-IID setting | | Dirichlet $\alpha = 0.1$ | | n-class (2) | | Dirichlet $\alpha = 0.1$ | | n-class (2) | |
| #Cluster | Methods | Accuracy | Macro-F1 | Accuracy | Macro-F1 | Accuracy | Macro-F1 | Accuracy | Macro-F1 |
| **1** | FedAvg | 85.90±0.46 | 54.52±2.66 | 86.17±0.25 | 44.88±1.24 | 25.62±3.47 | 11.38±2.02 | 24.30±3.53 | 8.56±0.64 |
| | FedProx | 86.03±0.58 | 54.69±3.32 | 86.47±0.23 | 44.89±1.38 | 25.72±3.29 | 11.14±1.49 | 24.19±2.45 | 8.69±0.74 |
| **5** | FedAvg+ | 86.12 | 61.07 | 86.5 | 45.39 | 25.71 | 12.45 | 24.83 | 8.74 |
| | FedProx+ | 86.39 | 56.56 | 86.15 | 45.43 | 25.58 | 12.43 | 25.88 | 8.55 |
| | IFCA | 90.13±6.81 | 68.47±5.23 | 91.54±5.04 | 72.30±5.32 | 47.21± 10.28 | 22.67±1.48 | 46.54±12.8 | 17.78±1.29 |
| | IFCA-CAM | 93.72±1.34 | 70.67±1.75 | 92.24±1.22 | 70.24±4.33 | 54.32±1.25 | 23.48±1.18 | 54.92±1.51 | 25.20±1.05 |
| | FeSEM | 91.51±2.90 | 73.78±9.88 | 91.83±1.24 | 71.05±8.63 | 54.30±4.58 | 24.78±6.01 | 55.55±4.83 | 32.80±4.18 |
| | FeSEM-CAM | **94.74±1.04** | **75.12±5.82** | **93.14±2.03** | **76.98±2.17** | **59.71±2.80** | **40.45±3.53** | **56.70±1.68** | **34.52±1.64** |
| **10** | FedAvg+ | 86.81 | 60.43 | 86.91 | 47.12 | 27.83 | 13.65 | 27.71 | 9.65 |
| | FedProx+ | 86.24 | 56.2 | 86.78 | 42.83 | 25.86 | 12.84 | 26.16 | 9.94 |
| | IFCA | 91.04±4.33 | 68.6±6.77 | 91.42±5.16 | 72.29±5.80 | 47.62±10.15 | 23.36±2.48 | 47.96±10.59 | 17.88±1.04 |
| | IFCA-CAM | **95.70±1.19** | 79.17±1.91 | 92.57±2.63 | 76.31±4.39 | 72.54±2.7 | 42.86±4.36 | 61.01±2.41 | 31.63±2.17 |
| | FeSEM | 93.3±2.0 | 80.47±11.05 | 93.75±1.53 | 79.39±6.57 | 67±1.57 | 31.69±8.52 | 63.64±6.51 | 42.97±6.08 |
| | FeSEM-CAM | 95.25±1.93 | **81.5±2.24** | **95.15±1.48** | **86.16±3.19** | **80.11±1.82** | **59.19±4.67** | **69.88±1.7** | **49.5±1.42** |

## 5.2 Visualization: CAM combats clustering collapse

Figures 1 and 3 demonstrate the effectiveness of applying CAM to mitigate clustering collapse in IFCA and FeSEM under both cluster-wise and client-wise non-IID scenarios, using the CIFAR-10 dataset with $K = 10$. Each color represents a cluster, and the X-axis represents the iteration rounds.

In the case of IFCA, we observe a severe clustering collapse issue in cluster-wise non-IID scenarios. A single cluster can encompass all clients in the client-wise non-IID setting and up to $50\%$ of clients in the cluster non-IID setting. Furthermore, the clustering remains unstable throughout the process. However, when CAM is applied in IFCA-CAM, it quickly identifies some clustering structures within a few rounds, and this structure closely approximates the ground truth.

As for FeSEM, while the phenomenon of clustering collapse is not as pronounced, a single cluster can still dominate up to $25\%$ of all clients if there are no outliers. CAM can expedite the clustering convergence, sometimes achieving it in just one round. Moreover, under client-wise non-IID settings, the application of CAM results in lower variance and more uniform cluster size. In the case of cluster-wise non-IID settings, FeSEM-CAM can easily identify the ground truth.

## 6 Conclusions

We propose a novel structured FL model "clustered additive modeling (CAM)" and an efficient FL algorithmic framework Fed-CAM to address non-IID FL challenges with clustering structure. CAM is a general mode-agnostic tool that can improve various existing non-IID FL methods. It can capture more general non-IID structures with global knowledge sharing among clients than clustered FL and overcome several weaknesses such as clustering collapse, vulnerability to cluster imbalance/initialization, etc. Theoretically, Fed-CAM is capable of achieving an asymptotic convergence rate of $O(1/\sqrt{T})$. Extensive experiments show that CAM brings substantial improvement to existing clustered FL methods, improves cluster balance, and effectively mitigates clustering collapse.

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
