# OpenReview forum: "Structured Federated Learning through Clustered Additive Modeling"
_NeurIPS.cc/2023/Conference — NeurIPS 2023 poster_

### Official Review · Reviewer_VZR1 · 2023-06-09

**Soundness:** 3 good
**Presentation:** 3 good
**Contribution:** 3 good
**Rating:** 8
**Confidence:** 5

**Summary:**

In this paper, the authors study heterogeneous federated learning, an important problem in federated learning, where the goal is to leverage the collective intelligence of multiple clients with diverse data distributions, features, or models to train a global model that can generalize well across all clients' data. In particular, the authors base their method on clustered federated learning and propose Clustered additive modeling to deal with the clustering collapse and dynamically changing models. The proposed method involves training a shared global model on top of cluster-specific models, and prediction results are obtained by combining the outputs from both the global model and the associated cluster model. Empirical results on two well-known datasets show the effectiveness of the proposed method.

**Strengths:**

Strength
1. The paper writing is clear and easy to follow
2. The empirical performance seems promising

**Weaknesses:**

Weakness:
1. Comparison with other methods under non-iid data setting needs to be improved.
2. Ablation study on structural clustering needs to be improved.
3. Fed-CAM introduces several hyper-parameters, including the number of rounds for warm-up, and the number of cluster size K.

**Questions:**

Questions for authors:
1. It is unclear which part of the structural clustering leads to the final good performance, as there are several hyper-parameters and assumptions in cluster generation. For example,
	1) How does the number of rounds for warm-up affect the quality?
	2) Can we dynamically adjust the cluster size K based on some coarse-to-fine heuristic?
	3) Can the proposed method be generalized to handle graph data?

2. More existing methods could be compared under more non-iid settings as well as iid settings. For example,
	1) more Distribution-based label imbalance setting, feature skew case, and label & feature skew case
	2) comparison with knowledge transfer-based methods

3. The authors do not include sufficient discussions about the communication and computational cost, which are suggested to be included.

**Limitations:**

Clustering-based FL approaches have a common limitation that is how to determine the cluster size. Some analysis on this would be desirable in the revision.

---

> ### Author Rebuttal · Authors · 2023-08-10
>
> Thank you for your positive feedback regarding our 'clear writing' and 'promising empirical performance.' Below, we provide our responses to the weaknesses and questions you mentioned in relation to this paper.
>
> ***Impact of warmup rounds***
> Please refer to the general response.
>
> ***Extra cost of CAM***
> Please refer to the general response.
>
> ***Chosen of Cluster Size $K$***
> The hyperparameter of cluster size $K$ is not introduced by CAM. It is from the original clustered FL methods. On one side, experiments of $K=5,10$ have been conducted under the ground truth of $K=10$. On the other side, While CAM can be an add-on to improve the performance of existing clustered FL methods, chosen of cluster size $K$ is also an independent branch to improve the performance. By the coarse-to-fine heuristic, we can set criteria, such as the loss difference between the local model and cluster model, to determine whether some clients need to be separated from a cluster or more clusters are needed.
>
> ***More Non-IID Settings, Datasets and Baselines***
> We have compared the baselines under four non-IID settings, including label imbalance and label skew, and will involve feature skew in future works.
> CAM can be generalized easily to existing FL methods handling graph data as an add-on. And we will involve more knowledge transfer-based methods in future works.
>
> We have tried to address most if not all concerns. Please let us know if you have any further questions. Thank you!
>
> Best,
> Authors

---

> > ### Comment · Reviewer_VZR1 · 2023-08-11
> >
> > Thanks for the clarificaiton. I am satisfied with the responses and revisions.

---

> > > ### Author Response · Authors · 2023-08-11
> > >
> > > Thank you very much for confirming our work and raising the score!

---

### Official Review · Reviewer_HLnM · 2023-06-25

**Soundness:** 2 fair
**Presentation:** 3 good
**Contribution:** 2 fair
**Rating:** 3
**Confidence:** 5

**Summary:**

This paper studied the problem of clustered federated learning. It analyzed the limitations of existing clustered FL algorithms, including clustering collapse and missing shared knowledge among clusters. Then this paper introduced a novel Clustered Additive Modeling (CAM) framework for learning both a globally shared model and cluster-wise models. A Fed-CAM algorithm was proposed to train CAM model parameters iteratively. The experimental results showed that CAM outperformed existing clustered FL algorithms in prediction accuracy.

#######################################
I appreciate the detailed response from the authors. However, my major concerns regarding the overall contributions still remain. Therefore, I keep my scores unchanged.

**Strengths:**

The strengths of this paper are summarized below.
(1) It proposed a novel Clustered Additive Modeling (CAM) framework for clustered federated learning. Compared to previous algorithms, CAM leveraged a global model to discover the shared knowledge among clusters.

(2) It demonstrated the flexibility of CAM framework by incorporating CAM with two popular clustered FL methods, thus leading to IFCA-CAM and FeSEM-CAM models. Then it provided a unified Fed-CAM optimization method for iteratively updating the clustering and global model parameters.

(3) Experimental results on both cluster and client-wise non-IID settings showed that compared to vanilla IFCA and FeSEM, IFCA-CAM and FeSEM-CAM could achieve better prediction performance in local clients.

**Weaknesses:**

(1) The rationale behind the proposed CAM framework lacks sufficient justification. As illustrated in the introduction section, existing cluttered FL algorithms suffers from clustering collapse, fragility to outliers, and sensitivity to initialization. However, the techniques regarding how CAM addresses these limitations are not explained.
(1-1) CAM introduced a global model to mitigate the lustering collapse. As shown in Subsection 3.3, it requires that global and cluster-wise models learn complementary knowledge. One concern is whether the global knowledge shared by all clients always exists in any non-IID FL settings. Furthermore, line 49 states that “different tasks or domains can benefit from sharing low-level or partial representations”. A natural question is whether it is more reasonable to learn partial representations (e.g., common low-level model parameters in FedRep [24]) for discovering the globally shared knowledge among clusters. In addition, the sensitivity of complementarity relationship between global and cluster-wise models can be analyzed with respect to warm-up strategies.
(1-2) This paper stated that the outlier clients have less impact on the global model, thus enabling the less vulnerability of CAM. But the outliers might have less common knowledge with other clients/clusters. Besides, based on the update rules in Eqs. (9)(14), the outliers can also dominate the training of global model, if they have a significantly large number of training samples.
(1-3) It is more convincing to quantitively evaluate the sensitivity of CAM to initialization, e.g., showing the clustering results with different random initializations in the experiments.

(2) The optimization of IFCA-CAM and FeSEM-CAM is confusing. Algorithm 1 updates the model parameters \theta_i and \theta_i^0 (lines 5-6) separately. This might lead to sub-optimal solutions of local objective functions, compared to optimizing \theta_i and \theta_i^0 at each training epoch simultaneously. In addition, it is much more efficient to update \theta_i and \theta_i^0 together within E training epochs.

(3) Though this paper focuses on clustered FL scenarios, it is better to compare the proposed CAM framework with other related FL frameworks. This is because learning the shared knowledge with a global model is a common strategy for personalized FL. For example, when each client is considered as a single cluster, the objective function of CAM is similar to [ref 1] and [ref 2]. In addition, the state-of-the-art pFL baselines based on feature sharing can be included to validate the performance of CAM.

(4) The convergence analysis of CAM is provided based on the assumption that clustering remains stable during training. But the results in subsection 5.2 might not validate the clustering stability. This is because Figures 1&3 provides the clustering results using only cluster size. It is not guarantee that clients in each cluster remains the same during training. More quantitively evaluations could be provided to verify the quality and stability of clustering in CAM.

[ref 1] Deng, Yuyang, Mohammad Mahdi Kamani, and Mehrdad Mahdavi. "Adaptive personalized federated learning." arXiv preprint arXiv:2003.13461 (2020).
[ref 2] Mansour, Yishay, Mehryar Mohri, Jae Ro, and Ananda Theertha Suresh. "Three approaches for personalization with applications to federated learning." arXiv preprint arXiv:2002.10619 (2020).

**Questions:**

(1) It shows that CAM tends to learn balanced clusters and determine the number of clusters automatically. Thus, CAM might perform well when the ground-truth of client clustering is balanced. But when the ground-truth of client clustering is highly skewed, would the balanced clusters learned by CAM result in sub-optimal solutions?

(2) Why is there a constant 1/m in Eq. (9) for updating global model parameters?

(3) In Eq. (13), what is the notation “\eta”? If it denotes the learning rate, why is it only applied to the first term of Eq. (12) with classification loss?

(4) In line 176, should it be “Eq. (13)” instead of “Eq. (3)” regarding the K-means regularization term?

(5) Why do IFCA-CAM and FeSEM-CAM use different warmup strategies? Is the selection of warmup strategy related to the clustering methods (e.g., minimum loss values in IFCA, K-means over parameters in FeSEM) in FL?

(6) What is FedAlt in line 217?

(7) In FL system setting, would the hyper-parameter \lambda be selected based on the best performance over the validation set? If so, how are the train/validation/test sets splits in the experiments?

**Limitations:**

Not discussed.

---

> ### Author Rebuttal · Authors · 2023-08-10
>
> Thanks for your suggestions and positive comments regarding our 'novel framework,' 'flexible framework,' and 'better empirical performance.' Below, we provide responses to the weaknesses and questions in your comments. Should you have additional questions, we are readily available for further discussion.
>
>
> ### Weakness Clarifications
> ***W1 (1-1) One concern is whether the global knowledge shared by all clients always exists in any non-IID FL settings?***
> As a fundamental assumption of most FL methods, we assume that there always exists some shared global knowledge. This holds true in practice since FedAvg usually sets up a reasonably good baseline. Most non-IID FL works address non-IID challenges by finetuning a personalised model based on a global model. The raised concern is out of the scope of FL research.
>
> ***W1 (1-2) the outliers can also dominate the training of global model, if they have a significantly large number of training samples***
> Our method CAM can handle outliers as long as they are the minority of the training set. If the outliers become the majority, they are not outliers anymore.
>
> ***W1 (1-3) It is more convincing to quantitively evaluate the sensitivity of CAM to initialization, e.g., showing the clustering results with different random initializations in the experiments.***
> We already reported the results with different random initializations in Table 1-2: the "Test results (means$\pm$std)" are based on 5 random seeds. Moreover, as demonstrated by the low stand deviation, integrating CAM with existing clustered FL methods (e.g. IFCA and FeSEM) can enhance the stability. We will further clarify it in the next version.
>
>
> ***W2 Why warmup and alternative optimization?***
> Both our convergence analysis and experiments demonstrate CAM's advantages over existing methods when using the warmup strategy and alternative optimization. In Fed-CAM algorithm, $\theta_i$ and $\theta_i^0$ are two different models that need to be optimized separately in parallel. They contribute to cluster-specific models and global models respectively. Updating $\theta_i$ and $\theta_i^0$ together without warmup degrades the performance as shown in Table 1 of the attached PDF in the general response. Moreover, it doubled the parameters to be trained at the same time, which is inefficient.
>
> ***W3 Similarity to APFL and MAPPER?***
> The methods of APFL [ref1] and MAPPER [ref2] interpolate between global and local model parameters, while CAM adds the outputs of global and cluster models as the final predictions. These are very different model architectures leading to different learning algorithms.
>
> We compared to pFL methods such as finetuning in Section 2.2 of Supplementary Material and we will add more in the next version.
>
> ***W4 Clustering stability analysis***
> The clustering converges fast as shown in Fig 1 of the attached PDF in the general response: Clients belonging to each cluster remain the same after a few global rounds.
>
> ### Question Clarifications
> ***Highly-skewed clustering analysis***
> Fig 2-3 of the attached PDF in the general response show that CAM still performs well under highly-skewed clustering.
>
> ***Warmup strategy***
> IFCA and FeSEM need different warmup strategies because they are initialized in different ways. The rule of thumb is to train the uninitialized models among the global model $\Theta_g$ and cluster models $\Theta_{1:K}$ by warmup stage. For example, IFCA initializes $K$ cluster models so IFCA-CAM's warmup needs to prepare a good global model. In contrast, for FeSEM initializing with $m$ client models whose aggregation already defines a global model, FeSEM-CAM's warmup should prepare $m$ client models for a better clustering.
>
> ***Dataset partition and chosen of $\lambda$***
> We split each client's local dataset into disjoint training, validation and test sets with a ratio of $6:2:2$ and tune $\lambda$ based on the validation set performance.
>
> ***Typos***
> We will correct typos raised in Questions 2, 3, 4, and 6. FedAlt was proposed in the same paper as FedSim.
>
> ***Limitations***
> We have discussed the limitations of our methods in the later paragraph of Introduction.
>
> [ref 1] Deng, Yuyang, Mohammad Mahdi Kamani, and Mehrdad Mahdavi. "Adaptive personalized federated learning." arXiv preprint arXiv:2003.13461 (2020).
>
> [ref 2] Mansour, Yishay, Mehryar Mohri, Jae Ro, and Ananda Theertha Suresh. "Three approaches for personalization with applications to federated learning." arXiv preprint arXiv:2002.10619 (2020).
>
> Please let us know if you have any further questions. If our responses have sufficiently addressed your concerns, it would be great if you are willing to reconsider your score kindly. Thank you!
>
> Best,
> Authors

---

> > ### Comment · Reviewer_HLnM · 2023-08-14
> > **Thank you for your response!**
> >
> > I appreciate the detailed response from the authors. However, my major concerns still remain.
> >
> > First of all, I respectfully disagree with "W1-1: The raised concern is out of the scope of FL research". When you apply a FL algorithm in practice, there's never guarantee that there's global knowledge shared by all clients, especially in the cross-device scenario, or when there are malicious clients.
> >
> > Second, for W1-2, there might be very few outlier clients, but each of them could have a large number of examples.
> >
> > Third, overall, this seems like "yet another" FL algorithm. The idea of combining the global model and the cluster-wise models seems interesting. However, the lack of theoretical insights significantly affects the contributions. Yes, the authors provided proof regarding the convergence. But another major theoretical aspect (arguably maybe more important) not discussed in the paper is when (under what conditions) the proposed algorithm would perform better than clustered FL, not just demonstrated by empirical results. Without this kind of theoretical results, I would consider the overall contributions underwhelming.

---

> > > ### Author Response · Authors · 2023-08-14
> > >
> > > Thanks for your reply! We would like to further address your concerns. You asked good questions, but they are not the problem studied in our paper. It is not realistic to completely solve all these problems in one paper. We focus on clustered federated learning and especially the clustering collapse problem, not how to deal with malicious/outlier clients or whether to apply federated learning (or global knowledge sharing). Although our method can mitigate these two problems as its byproducts, they are not what we mainly aim to solve in this paper. More detailed discussion:
> > >
> > > - It is not "Federated Learning" but "local SGD" if no global knowledge can be shared across clients. In practice, it is possible that some (malicious) clients do not share any knowledge with the others. Our model is able to capture this case by learning a local cluster model for such clients whose output dominates the global model output in the additive prediction, so they share nearly zero knowledge with others. Moreover, existing methods can lower these clients' importance or remove them entirely [ref 1-3] and they can be seamlessly integrated into our CAM framework.
> > >
> > > - Outlier clients with a large number of examples can be addressed by applying equal weights to all clients, which is commonly used in popular FL methods [ref 4-6] with thousands of citations. Even without using equal weights and when the global model is dominated by such an outlier, our method can learn cluster models for other non-outlier clients and let their outputs dominate the global model output in the additive prediction, hence robust to such outliers.
> > >
> > > - We have clarified that our method is fundamentally different from APFL or MAPPER or other FL methods: they compute a sum of global/local model parameters while our model computes a sum of global/local model outputs. Besides the convergence analysis, in our introduction and experiment sections, we provided a thorough analysis and empirical evidence explaining why adding CAM to existing clustered FL is critical to overcome their current shortcomings.
> > >
> > > [ref 1] Sun, Ziteng, et al. "Can you really backdoor federated learning?." arXiv preprint arXiv:1911.07963 (2019).
> > >
> > > [ref 2] Bagdasaryan, Eugene, et al. "How to backdoor federated learning." International conference on artificial intelligence and statistics. PMLR, 2020.
> > >
> > > [ref 3] Wang, Hongyi, et al. "Attack of the tails: Yes, you really can backdoor federated learning." Advances in Neural Information Processing Systems 33 (2020): 16070-16084.
> > >
> > > [ref 4] Li, Tian, et al. "Federated optimization in heterogeneous networks." Proceedings of Machine learning and systems 2 (2020): 429-450.
> > >
> > > [ref 5] Karimireddy, Sai Praneeth, et al. "Scaffold: Stochastic controlled averaging for federated learning." International conference on machine learning. PMLR, 2020.
> > >
> > > [ref 6] Ghosh, Avishek, et al. "An efficient framework for clustered federated learning." Advances in Neural Information Processing Systems 33 (2020): 19586-19597.

---

### Official Review · Reviewer_WQSD · 2023-07-08

**Soundness:** 4 excellent
**Presentation:** 4 excellent
**Contribution:** 3 good
**Rating:** 8
**Confidence:** 5

**Summary:**

This paper proposes a new federated learning with a specified structure for the prediction produced for each client, called "clustered additive modelling", which adds the prediction of a global model to the prediction of the model trained on a local cluster of clients. It is a modification to existing clustered federated learning methods and fixes several critical issues of these methods such as clustering collapse, fragility to outlier clients, and sensitivity to initialization. Some of the issues are fundamental and widely exist in many existing methods. The paper shows that the proposed CAM can effectively address these issues in different clustered FL methods under different non-IID settings, consistently leading to a promising improvement.

**Strengths:**

1. This tackles a fundamental problem in clustered FL, i.e., clustering collapse. This problem might be widely observed but has not been well investigated and effectively solved before. Hence, I believe this work can be helpful and have a broad impact on FL problems studying statistical heterogeneity and structures among clients.
2. The motivation of this paper is clear. The solution provided by clustered additive modelling is principal, easy to understand, effective across different settings, and generalizable to various existing FL methods.
3. The proposed algorithms in Section 3.1 and 3.2 are intuitive and convincing with detailed explanations per step and theoretical guarantee (convergence analysis). I appreciate the authors for providing two examples of CAM algorithms following the same high-level idea.
4. The experiments conducted on 2 datasets x 4 non-IID settings show great advantages of the proposed method over existing FL and clustered FL methods. The analysis of clustering collapse also provides a nice explanation and empirical evidence for the improvement.
5. The code is provided for high reproducibility.


**Weaknesses:**

1. Compared to existing clustered FL methods, the proposed method needs to train an additional model. How much extra computation does it require? It would be helpful to provide the computation complexity of CAM.
2. While other parts of the paper are easy to understand, the convergence analysis part is limited in length and left many details in the appendix. Can you elaborate a little bit more on the proof idea?
3. The comparison and experiments on each dataset are thorough to me but the empirical results would be stronger if experimenting on more datasets.


**Questions:**

1. Can you provide more details and some examples of the cluster-wise non-IID setting?
2. What is the cost of the warmup stage? How is the performance affected by reducing the warmup rounds?
Overall, I think this is a solid paper that successfully addressed a widely existing fundamental problem in FL that has not been investigated before. The motivation and proposed idea are principal. Most parts of the paper are clear. The analysis and experiments are convincing. Experiments on more datasets and more guidance on the theoretical analysis can further strengthen the draft.


**Limitations:**

Yes. The authors have addressed the limitations.

---

> ### Author Rebuttal · Authors · 2023-08-10
>
> Thank you for your positive feedback regarding our 'clear motivation tackling a fundamental problem in clustered FL,' 'intuitive and convincing algorithms,' 'nice analysis for clustering collapse,' and 'high reproducibility.' Below, we provide our responses to the weaknesses and questions you mentioned in relation to this paper.
>
>
> ### Weakness Clarifications
>
> ***Extra cost of CAM***
> Please refer to the general response.
>
> ***Convergence framework***
> We will add the main steps of convergence analysis into the main paper in future works. The proof framework of convergence analysis is to bound the error of one communication round first, then add $T$ rounds together. For the error of one communication round, we break it into three parts, local training, clustering and aggregation, and bound each separately.
>
> ***Experiments on more datasets***
> We will test CAM on more cases, such as graph and NLP in future works.
>
> ### Question Clarifications
>
> ***Examples of cluster-wise non-IID settings***
> In the real world, clustering is very popular and used everywhere. For example, the Mall Customer Segmentation Dataset includes customer information like age, income, and spending score. Clustering can help identify different customer segments based on their shopping behavior. The 20 Newsgroups Dataset contains newsgroup documents from 20 different newsgroups, covering a wide range of topics. Clustering can be used to group documents with similar content, such as sports, politics, science, etc.
>
> ***Impact of warmup rounds***
> Please refer to the general response.
>
> We have tried to address most if not all concerns. Please let us know if you have any further questions. Thank you!
>
> Best,
> Authors

---

### Official Review · Reviewer_dS6T · 2023-07-12

**Soundness:** 3 good
**Presentation:** 3 good
**Contribution:** 3 good
**Rating:** 6
**Confidence:** 4

**Summary:**

This paper proposes a novel clustered federated learning method using additive modeling to tackle the clustering collapse problem. The paper contributes to advance the research domain of clustered federated learning which is an important and practical solution to solve non-iid problem in federated settings.

**Strengths:**

1. The proposed method is simple yet effective. The overall solution is technique sounds. This work is very likely will become a new baseline of clustered FL.

2. The targeting problem is an inherent challenge of clustered FL. The Introduction section provides insight analysis of current clustered FL.

3. The overall flow description is sufficiently clear.

4. The provided theoritical analhysis is well suited to the clustered FL settings.

5. The claims are well supported by the experiments.

**Weaknesses:**

1. The readability of this paper could be improved. For example, there are many abbreviation names, e.g. CAM, FED-CAM, IFCA, FeSEM. The symbol system is a little bit complicated.

2. The paper discussed several insightful challenge of clustered FL. However, it is how the proposed method can solve these challenges.

3. Some description needs to be improved. For example, it is unclear how the Figure 1 and 2 link to the Introduction section. Moreover, in Introduction section, there are many symbols without a clear definition.

4. This method may require storing multiple models locally (cluster-level model, global model, and local model), which could increase costs. It would be better to compare and discuss with other methods to explore potentially more favorable alternatives.

**Questions:**

1. In Eq.1, why you use two symbols h(.) and f(.) to represent global model and cluster-specific model. Are they using different model architecture?

2. In Eq. 2, why is the additive results is Y rather than 2Y?

3. Please discuss how the proposed method can solve the mentioned challenges in Introduction section.

4. Is it necessary to have a comparison experiments with some personalized FL method, e.g. FedAvg + Finetuning?

5. In Ensemble FL, whatis the different part of K times experiments?

6. How about to add a parameter to adjust the importance weight between two models h and f? Say h(.) + a*f(.).

7. In Notations (line 128), what does the symbol "0" represent in the context of the symbol $\theta_{1:m}^0$ ?

8. Is the performance of the global model or the cluster-level model compared during testing?

9. How many clusters obtained in training phase? Is it necessary to keep the same number of clusters as the CAM method for methods that need to specify the number of clusters?

**Limitations:**

Yes. The authors have addressed the limitations. There is no negative societal impact of this work.

---

> ### Author Rebuttal · Authors · 2023-08-10
>
> Thank you for your positive feedback regarding our 'simple-yet-effective method,' 'well-suited theoretical analysis,' and 'well-supported claims.' Below, we provide our responses to the weaknesses and questions you mentioned in relation to this paper.
>
> ### Weakness Clarifications
>
> ***How does CAM solve the mentioned challenges?***
> - CAM solves the mentioned challenges by introducing a global model $\Theta_g$ and adding its output to the output of the cluster model $\theta_{c(i)}$ each client $i$ belonging to. The final prediction is the sum of the global model output and cluster model output $y = h(x; \Theta_g) + f(x; \theta_{c(i)})$. This removes any globally shared components from cluster models and thus prevents one cluster model from capturing the globally shared knowledge, which leads to clustering collapse.
>
> - Moreover, this also removes the conflicts between client clustering and global knowledge sharing, where the former moves clusters models away from each other and thus harms global sharing, because CAM optimizes the two objectives separately on two groups of model parameters. Therefore, the global model solely focuses on capturing the globally sharable knowledge, while the cluster models focus on intra-cluster shared but inter-cluster separable knowledge.
>
> - Furthermore, CAM improves robustness against outliers, which primarily impact $\Theta_{1:K}$ but exert minimal influence on the global model $\Theta_g$. Notably, the interplay between $\Theta_{1:K}$ and $\Theta_g$ renders CAM less susceptible to initial cluster assignments' variations, as adjustments to $\Theta_g$ induce changes in the clustering scheme.
>
> ***Extra cost of CAM***
> Please refer to the general response.
>
> ***Readability***
> For readability-related Weaknesses 1 and 3, most of them can be found in the introduction section. The full name of CAM can be found in the abstract, while Fed-CAM denotes "Federated Learning with CAM". In the introduction, IFCA and FeSEM have been cited in Line 84. Fig 1 and 2 have been discussed in Lines 79 and 89, respectively.
>
>
> ### Question Clarifications
> ***In Eq.1, why you use two symbols $h(.)$ and $f(.)$ to represent global model and cluster-specific model. Are they using different model architecture?***
>
> Yes. The global model can have a different architecture as the cluster-specific model as long as their outputs are addable.
>
> ***In Eq. 2, why is the additive results is $Y$ rather than $2Y$?***
>
> The additive results will be processed by softmax before being compared with $Y$. Moreover, the models are learnable so as the scales of their outputs. So only $Y$ is needed.
>
> ***Is it necessary to have a comparison experiments with some personalized FL method, e.g. FedAvg + Finetuning?***
>
> Yes, we have included personalized FL methods as baselines in Section 2.2 of the supplementary material, in which Fedavg+Finetuning and some other personalized FL methods have been compared.
>
> ***In Ensemble FL, whatis the different part of K times experiments?***
>
> In Ensemble FL, the different part of $K$ times is the initialization.
>
> ***How about to add a parameter to adjust the importance weight between two models h and f? Say $h(.) + a*f(.)$.***
>
> It is not necessary because the importance weights can be automatically learned by the models, i.e., the model parameters can control the scales of the model output. For inference, an extra importance weight to makes it inconsistent with the training.
>
> ***In Notations (line 128), what does the symbol "0" represent in the context of the symbol $\theta^0_{1:m}$?***
>
> Both $\theta^0_{1:m}$ and $\theta_{1:m}$ are local models but they are different: $\theta^0_{1:m}$ are trained and aggregated for the global model optimization, while $\theta_{1:m}$ are trained and aggregated for the cluster model optimization.
>
> ***Is the performance of the global model or the cluster-level model compared during testing?***
>
> Yes. Global models such as FedAvg and FedProx and cluster-level models such as IFCA and FeSEM have been compared during testing in Tables 1-2 of the main paper.
>
> ***How many clusters obtained in training phase? Is it necessary to keep the same number of clusters as the CAM method for methods that need to specify the number of clusters?***
>
> As shown in the first column of Tables 1-2, the chosen number of clusters is $\{1,5,10\}$, while the ground truth is $10$. And for the second question, YES, the same number of clusters is kept for comparison.
>
> Please let us know if you have any further questions. If our responses have sufficiently addressed your concerns, it would be great if you are willing to reconsider your score kindly. Thank you!
>
> Best,
> Authors

---

> > ### Comment · Reviewer_dS6T · 2023-08-10
> >
> > Thanks for the clarification. Despite some heuristic aspects that may affect the robustness, the proposed method might be a viable solution to handle FL by introducing structural knowledge.

---

> > > ### Author Response · Authors · 2023-08-11
> > >
> > > Thanks for confirming our clarifications. Please let us know if you have further concerns. If most of your concerns have been addressed, it would be great if you are willing to reconsider your score kindly. Thanks!

---

### Author Rebuttal · Authors · 2023-08-10

## General response
Our proposed method mentioned warmup stage can improve the performance of clustered FL methods, and the framework is flexible to be integrated with existing FL methods. Reviewers show significant interest in these two points by asking for more details and support to these two points. We respond to them by adding two more analyses as below.

***Impact of Warmup Rounds***
As shown in Table 1 below, we gradually increase the rounds of the warmup stage (from 0 to 50) while keeping the total budget of rounds to 100 (warmup + training), considering the limited capacity of computation and communication for local devices in FL. The best performance is achieved when the warmup rounds are set to 20. However, the performance shows minimal variation when the number is set to 10, 20, 30, or 40. It demonstrates that the performance is stable when we choose warmup rounds in the area from 10 to 40. The choice of warmup round numbers exhibits low sensitivity, like on the parameter plateau.

Notably, with no warmup rounds, performance is substantially decreased due to the impact of worse-performed initial candidates of the FL system. Similarly, when the warmup rounds are increased to 50, indicating insufficient training, the performance will drop accordingly. We need to ensure there are enough training rounds with a proper number of warmup rounds.

In summary, a few warmup rounds can improve the stability of FL optimization and accuracy-related performance. Given a proper area, choosing warmup rounds is low sensitivity to performance.

***Extra Cost of integrating proposed CAM framework to existing FL methods***
For simplicity, we use 'FedAvg' as the measuring unit or benchmark for the cost of storage, communication and computation on local devices. In general, CAM will bring one extra 'FedAvg' cost to the existing FL methods every communication round.

As for IFCA [ref 1], which needs to transmit $K$ cluster-specific models to each client to compute the clustering, applying our proposed CAM framework with IFCA, we need to transmit $K$ cluster models and one extra global model to the clients, that is $K+1$ models in total. The communication cost and storage cost are listed in Table 1. Moreover, the warmup stage only incurs one 'FedAvg' cost. Therefore, integrating CAM can even reduce the overall cost by increasing the number of warmup rounds.

Lastly, considering the tradeoff between performance and cost, we choose 30 warmup rounds out of 100 as the default experiment setting.

**Table 1**: Ablation study of warmup round numbers for performance and cost using "FedAvg" as the measuring unit (Other settings: CIFAR-10 dataset, IFCA, client-wise non-IID with Dirichlet distribution $\alpha = 0.1$, Cluster number $K=10$). For more details, please refer to Table 1 of the attached PDF.

| Baseline | # Warmup + Training |  Accuracy/% |  Macro-F1/% | Storage cost/'FedAvg' | Communication cost/'FedAvg' | Computation cost/'FedAvg' |
|:--------:|:-------------------:|:---------:|:---------:|:------------:|:------------------:|:----------------:|
|   IFCA   |        0+100        |   47.62   |   23.36   |      **10x**     |         10x        |        10x       |
| IFCA-CAM |        0+100        |   63.75   |   32.17   |      11x     |         11x        |        11x       |
| IFCA-CAM |        10+90        |   72.69   |   41.24   |      11x     |         10x        |        10x       |
| IFCA-CAM |        20+80        | **73.83** | **44.72** |      11x     |         9x         |        9x        |
| IFCA-CAM |      **30+70**      |   72.54   |   42.86   |      11x     |         8x         |        8x        |
| IFCA-CAM |        40+60        |   72.98   |   42.20   |      11x     |         7x         |        7x        |
| IFCA-CAM |        50+50        |   65.74   |   26.63   |      11x     |       **6x**       |      **6x**      |


[ref 1] Avishek Ghosh, Jichan Chung, Dong Yin, and Kannan Ramchandran. An efficient framework for clustered federated learning. Advances in Neural Information Processing Systems, 33:19586–19597, 2020.

---

### Decision · Program_Chairs · 2023-09-21

**Decision:**

Accept (poster)

**Comment:**

This paper introduces a novel approach to federated learning, termed "clustered additive modeling" (CAM), which incorporates a structured prediction framework for individual clients. CAM combines the predictions of a global model with those of locally trained models within specific client clusters. This modification represents an enhancement over existing clustered federated learning techniques, as it successfully mitigates several critical challenges commonly associated with these methods. The paper demonstrates that CAM effectively addresses these issues across various clustered federated learning methods operating in diverse non-IID (non-identically distributed) scenarios, consistently yielding promising improvements.

This paper addresses a critical issue in clustered federated learning, known as "clustering collapse," which has been widely observed but not thoroughly explored or resolved previously. The paper's contribution is expected to be highly beneficial and influential in the field of federated learning, particularly in studying statistical heterogeneity and client structures. The motivation behind this research is clear, and the proposed solution through clustered additive modeling is both fundamental and effective. Nevertheless, one reviewer has concerns on the performance of the algorithms when there exist significant amount of outliers in the network, which potentially break the assumptions in the paper. While this scenario can be potentially considered as future works, I would suggest the authors add discussions in the method limitation section. Overall, I recommend acceptance of this paper.